# Vascular Extracellular Matrix in Atherosclerosis

**DOI:** 10.3390/ijms252212017

**Published:** 2024-11-08

**Authors:** Alessia Di Nubila, Giovanna Dilella, Rosa Simone, Silvia S. Barbieri

**Affiliations:** Unit of Brain-Heart Axis: Cellular and Molecular Mechanisms, Centro Cardiologico Monzino IRCCS, via Parea 4, 20138 Milan, Italy; alessia.dinubila@cardiologicomonzino.it (A.D.N.); giovanna.dilella@cardiologicomonzino.it (G.D.); rosa.simone01@universitadipavia.it (R.S.)

**Keywords:** extracellular matrix, vascular tissue, atherosclerosis, collagen, elastin, fibronectin, proteoglycans

## Abstract

The extracellular matrix (ECM) plays a central role in the structural integrity and functionality of the cardiovascular system. Moreover, the ECM is involved in atherosclerotic plaque formation and stability. In fact, ECM remodeling affects plaque stability, cellular migration, and inflammatory responses. Collagens, fibronectin, laminin, elastin, and proteoglycans are crucial proteins during atherosclerosis development. This dynamic remodeling is driven by proteolytic enzymes such as matrix metalloproteinases (MMPs), cathepsins, and serine proteases. Exploring and investigating ECM dynamics is an important step to designing innovative therapeutic strategies targeting ECM remodeling mechanisms, thus offering significant advantages in the management of cardiovascular diseases. This review illustrates the structure and role of vascular ECM, presenting a new perspective on ECM remodeling and its potential as a therapeutic target in atherosclerosis treatments.

## 1. Introduction

The extracellular matrix (ECM) is a fundamental component of the cardiovascular system and plays a crucial role in maintaining the structural integrity, functionality, and overall health of blood vessels and the heart [1]. The ECM is critical for mechanotransduction processes as it provides the necessary tensile strength to withstand the high pressure exerted by blood flow, ensuring that blood vessels and cardiac tissue maintain their shape and withstand mechanical stresses [2]. It gives arteries elasticity so that they can expand and contract with every heartbeat. This elasticity is essential for maintaining a smooth and even blood flow, adapting to changes in blood volume, and regulating blood pressure [3]. In addition to providing a physical scaffold, the ECM is actively involved in various cellular and molecular processes that are important for cardiovascular homeostasis and response to injury. It promotes cell adhesion, migration, proliferation and differentiation, and influences phenotypic cell modulation, which affects inflammatory and repair processes [4]. By maintaining the integrity of the endothelial layer, the ECM ensures adequate blood flow and blood pressure, and reduces the risk of blood clot formation. Moreover, ECM components contribute to the formation and stability of atherosclerotic plaques [5]. In this review, we will discuss the structure and function of the vascular ECM, focusing on its role and remodeling in atherosclerosis. Finally, the ECM as a potential therapeutic target for atherosclerosis will be considered (Figure 1).

## 2. Structure and Functions of the ECM

Traditionally, the ECM has been viewed primarily as a passive structural scaffold that maintains the architecture and mechanical integrity of tissue. The ECM represents a complex network of fibrillar and non-fibrillar components that have both signaling and structural functions in tissue organisation, remodeling, and the regulation of cellular processes [6,7]. The main components of the ECM include collagens, elastin, fibronectin, laminin, and proteoglycans, which together create a microenvironment that regulates cell adhesion, migration, proliferation, and differentiation [8]. These components interact to provide the mechanical properties and biological functions necessary to maintain vascular integrity [9].

However, the composition of the ECM is tissue-specific and changes within a single tissue during development, homeostasis, and disease. Different vessels have a tissue-specific morphology and ECM composition with specific characteristics: large elastic arteries (aorta and great vessels), muscular arteries and arterioles, and capillaries as well as venules and veins [1]. The structural proteins of the arterial ECM provide a uniform distribution of loads to protect against vessel rupture [10]. The arterial wall consists of three layers: the tunica intima, tunica media, and adventitia (Figure 2).

The tunica intima (Table 1), characterized by a proteoglycan-rich matrix (i.e., versican, hyaluronic acid, decorin), consists of a longitudinally arranged endothelial cell lining, a basement membrane, and a subendothelial cell connective tissue region with sporadic smooth muscle cells [11]. The basement membrane is a thin sheet-like ECM structure containing mainly laminin isoforms, collagen type IV, nidogens, perlecan, fibronectin (FN), and other molecules including growth factors (i.e., vascular endothelial growth factors (VEGFs)), matrix metalloproteinases (i.e., collagenases, gelatinases), the Adam (a disintegrin and metalloproteinase)/Adamts (a disintegrin and metalloproteinase with thrombospondin motifs) proteinase inhibitor, and metalloproteinase inhibitors (e.g., tissue inhibitors of metalloproteinases 3, TIMP-3) [6,12,13].

The tunica media (Table 1), the thickest layer, comprises lamellar units containing elastic lamellae and an interlamellar space with smooth muscle, elastin fibers, collagen (types I and III), and proteoglycans. This layer is highly cross-linked by lysyl oxidase (LOX) isoforms and transglutaminases to provide mechanical stability to the highly stressed vessel wall, thus exhibiting vasoconstrictor and vasodilator capabilities essential for the regulation of blood flow [14].

The outer layer, the tunica adventitia (Table 1), contains connective tissue with large bundles of type I collagen, fibroblasts, and endothelial cells that form a network of microvessels called vasa-vasora, and immune cells. Between the tunica intima and tunica media is an inner elastic lamina; in addition, the tunica media and the tunica adventitia are separated by an outer elastic lamina [15].

Like the arteries, the veins also have a three-layer structure (Figure 3). However, compared to arteries, veins have an overall thinner wall with a smaller tunica media and a thicker adventitial layer with a high collagen content. This is a direct consequence of the lower blood pressure in the veins [16].

Although the matrisome includes approximately 300 proteins [17], the mechanical properties of the vessel wall, such as high resilience, nonlinear elasticity, and low hysteresis, are mainly provided by three components: fibrillar collagens, elastic fibers, and large proteoglycans [18].

ECM fulfills several important functions that influence vascular development, homeostasis, and repair, as follows:

(a) Structural support. The ECM provides a scaffold that maintains the integrity of the vessel walls and ensures that they can withstand the mechanical stresses imposed by blood flow [19]. Important components of the ECM, such as collagen and elastin, are responsible for tensile strength and elasticity, respectively. Collagen fibers form a robust network that resists stretching and tearing, while elastin fibers allow blood vessels to stretch and retract, maintaining the dynamics of blood flow.

(b) Regulation of cell behavior. However, it is well known that the ECM is much more than just a static scaffold. It is an active participant in the regulation of cellular behavior and influences processes such as cell adhesion, proliferation, migration, and differentiation [20,21]. This regulation is mediated through interactions between ECM components and cell surface receptors, such as integrins [22,23].

(c) Barrier function. The basement membrane, a key component of the ECM, acts as a selective barrier, and regulates the exchange of molecules between the bloodstream and surrounding tissues. It also plays a role in maintaining endothelial cell polarity and vascular stability [24].

(d) Signal reservoir. The ECM stores growth factors, cytokines, and other signaling molecules, and modulates their availability and activity. This storage and controlled release are crucial for processes such as angiogenesis, inflammation, and tissue repair [25].

## 3. Role of ECM in Atherosclerosis

The ECM is not just a passive scaffold, but an active participant in the pathogenesis of atherosclerosis. It influences several key processes, including endothelial dysfunction, lipid retention, smooth muscle cell migration and proliferation, and plaque stability [26] (Figure 4).

### 3.1. Endothelial Dysfunction and Permeability

The first step in atherogenesis involves endothelial dysfunction, characterized by increased permeability, decreased nitric oxide (NO) production, and enhanced expression of adhesion molecules, such as vascular cell adhesion molecule-1 (VCAM-1) and intercellular adhesion molecule-1 (ICAM-1) [27,28]. The ECM, in particular the basement membrane, plays a critical role in maintaining endothelial cell function.

ECs can undergo endothelial-to-mesenchymal transition (EndMT), a dedifferentiation process in which they lose their endothelial properties (e.g., reduced expression of endothelial markers such as VE-cadherin and endothelial nitric oxide synthase (eNOS)) [29,30] and gain mesenchymal features (e.g., increased expression of mesenchymal markers such as alpha-smooth muscle actin, vimentin, and N-cadherin) [31]. During EndMT, the disruption of endothelial junctions allows high amounts of inflammatory cells, lipids, and immune complexes to enter the intima, which further exacerbates the disruption of ECM components such as laminin and collagen IV and promotes plaque growth [32,33]. In addition, the degradation of ECM components, particularly by matrix metalloproteinases (MMPs), can release matrix-bound growth factors, such as transforming growth factor-β (TGF-β), which promotes EndMT and enhances the fibrotic and inflammatory milieu within the plaque [34].

### 3.2. Lipid Retention and Oxidation

A crucial early event in atherogenesis is the retention of lipoproteins in the arterial wall [35]. Proteoglycans in the ECM interact closely with low-density lipoprotein (LDL) particles, particularly when the LDL is modified (e.g., oxidized or glycosylated) [36]. Biglycan and decorin, for example, bind to apoB-100 on LDL, and trap these lipoproteins in the arterial wall. Once trapped, these lipoproteins undergo oxidative modification, which triggers an inflammatory response and promotes the recruitment of monocytes that differentiate into macrophages and ingest modified LDL to become foam cells [37,38].

### 3.3. Smooth Muscle Cell Migration and Proliferation

Vascular smooth muscle cells (VSMCs) play a critical role in the pathogenesis of atherosclerosis. In response to injury or inflammatory stimuli, VSMCs migrate from the media to the intima, where they proliferate and secrete ECM components such as collagen and elastin. This process contributes to the formation of the fibrous cap that covers atherosclerotic plaques [39]. However, the phenotype of VSMCs changes during atherosclerosis. It transforms from a contractile to a synthetic phenotype, characterized by increased ECM production and decreased contractility [40]. This synthetic form is characterized by increased proliferation, migration, and reduced expression of contractile proteins, as well as increased ECM production [41]. Synthetic VSMCs produce ECM components such as collagen and elastin, which may contribute to plaque stabilization. However, their ability to secrete MMPs can degrade the ECM, destabilizing the plaque and making it susceptible to rupture [42].

Interestingly, changes in ECM stiffness due to collagen accumulation, elastin degradation, or other ECM components may influence the phenotypic transition of smooth muscle cells (SMCs) [43]. Increased ECM stiffness has been associated with the promotion of the synthetic phenotype of VSMCs, leading to excessive ECM production and contributing to plaque growth [43].

### 3.4. Inflammation and Immune Cell Recruitment

The ECM plays an active role in modulating the inflammatory response, and recruits immune cells to the atherosclerotic lesion [44]. ECM components, such as fibronectin and proteoglycans, can bind and sequester chemokines and cytokines, creating a local pro-inflammatory environment. Macrophages in particular play a decisive role in the progression of atherosclerosis. Depending on the local environment, they can polarize into either pro-inflammatory M1 macrophages or anti-inflammatory M2 macrophages [45,46]. M1 macrophages promote ECM degradation by secreting MMPs and inflammatory cytokines that weaken the plaque fibrous cap [47]. In contrast, M2 macrophages contribute to tissue repair and ECM deposition and support plaque stability [48]. Macrophages also undergo a phenotypic transition into foam cells when they uptake oxidized low-density lipoproteins (oxLDL) [49]. Foam cells contribute to the formation of the lipid-rich necrotic core of atherosclerotic plaques and secrete ECM-degrading enzymes, which further contribute to plaque vulnerability [50,51]. In addition, the degradation products of ECM proteins, generated by enzymes like MMPs, can act as damage-associated molecular patterns (DAMPs) that activate pattern recognition receptors (PRRs) on immune cells and further amplify the inflammatory response [52].

### 3.5. Pericytes and Adventitial Fibroblasts

Adventitial fibroblasts, which are located in the outermost layer of blood vessels, can transition into myofibroblasts during atherosclerosis [53]. Myofibroblasts are highly proliferative and contribute to the excessive deposition of the ECM, especially collagen, which may stabilize the plaque but also lead to fibrosis and vascular stiffness [54,55]. Pericytes, another type of mural cell, can undergo phenotypic changes in response to vascular injury or inflammation, transitioning into fibroblasts or even myofibroblasts, which contribute to ECM deposition and vascular remodeling [56].

### 3.6. Plaque Stability and Vulnerability

The balance between ECM synthesis and degradation is crucial for plaque stability. A stable plaque is typically characterized by a thick fibrous cap, rich in collagen, produced by VSMCs. Conversely, a vulnerable plaque has a thin fibrous cap and a large necrotic core, often due to increased ECM degradation by MMPs and reduced collagen synthesis [57]. The degradation of collagen and elastin weakens the fibrous cap and makes it more prone to rupture. Plaque rupture exposes the underlying thrombogenic material to the bloodstream, leading to thrombus formation and potentially fatal clinical events such as myocardial infarction or stroke [58].

## 4. Key ECM Proteins in Atherosclerotic Plaques

The ECM in the vessel wall undergoes extensive remodeling during atherosclerosis, affecting plaque stability, cell migration, and inflammatory responses. Understanding the role of ECM proteins in atherosclerosis is critical for the development of targeted therapies to mitigate the progression of this disease.

This section can be divided by subheadings. It should provide a concise and precise description of the experimental results, their interpretation, and the experimental conclusions that can be drawn.

### 4.1. Collagen

Collagens are key structural proteins in the ECM and are essential for the mechanical stability of atherosclerotic plaques [59]. There are at least 29 different forms of collagens [60], grouped according to their structural functions. They generate thick parallel bundles or fibrils (I, II, III, V, XI, XXIV, XXVII), associate with collagen fibrils (FACITs) (IX, XII, XIV, XVI, XIX, XX, XXI, XXII), form networks (IV, VIII, X), beaded filament (VI, XXVI, XXVIII) or anchoring fibril (VII), and transmembrane proteins (XIII, XVII, XXIII, XXV) [61,62]. All collagens have a triple helical structure consisting of three polypeptide chains [63], and play a crucial role in maintaining tissue strength, adhesion, and supporting cell migration [64]. Collagens I, III, and IV [65] are predominant in the vascular wall. Collagen IV is essential for basement membrane stability, providing structural support to endothelial cells and maintaining the integrity of the endothelial barrier, while collagen XVIII is involved in supporting its integrity [66]. Collagens I and III, found in the tunicae media and adventitia, provide tensile strength [67]; however, vessel injury can alter the balance between these two types of collagens [68,69]. Meanwhile, collagen IV is essential for basement membrane stability, endothelial cell support, and maintaining the integrity of the vascular barrier. Alteration in collagen IV can lead to increased permeability and leukocyte infiltration, inflammation, and plaque instability [70]. However, in advanced lesions, the balance between collagen synthesis and degradation is disrupted, leading to the thinning of the fibrous cap and increased risk of plaque rupture [58,71]. In the media and early atherosclerotic lesions, VSMCs are the main source of collagen I and III, which they synthesize in mechanical stress and inflammation [69,70]. As atherosclerotic plaques develop collagen, synthesis increases and eventually accounts for up to 60% of the total protein in the plaque matrix [71]. In the early stages of atherosclerosis, this increase in collagen helps to stabilize the plaque by forming a strong fibrous cap. In advanced lesions, however, the balance between collagen production and degradation is disturbed, leading to the thinning of the fibrous cap and higher risk of plaque rupture. In advanced plaques, collagen IV thickens and collagen V is highly expressed [71,72], while collagen degradation in the lipid core is accelerated [73]. These processes are regulated by factors such as TGF-β, which promotes collagen synthesis, and inflammatory cytokines, which increase the activity of MMPs that degrade collagen [74].

### 4.2. Elastin

Elastin, a resilient connective tissue protein, is formed by the multimerization and cross-linking of its monomers (68–74 kDa), the tropoelastins [72]. Tropoelastin is synthesized by SMCs in the media and by fibroblasts in the adventitia, and then undergoes cross-linking and fiber formation in the ECM under catalysis by lysyl oxidase and the helper proteins fibulin-4 or -5 [68]. Since there are numerous cross-links at lysine residues and hydrophobic domains account for almost 75% of the total protein composition, elastin is extremely hydrophobic [75]. In fact, the small non-polar amino acids glycine, leucine, valine, and proline are localized in hydrophobic domains, while the hydrophilic domains contain lyine–alanine and lysine–proline motifs [76]. The degree of elastin cross-linking is directly proportional to the tensile strength of the ECM in the arterial wall. Elastin provides vessels with the ability to stretch and retract in response to the hemodynamic forces generated by arterial pulsation and intravascular pressure [77]. Thick layers of elastic fibers characterize the internal elastic lamina at the intima-medial border in arteries and veins and the additional laminae in the medial layer of large arteries [16,78]. Over time, the amount of elastin decreases and the amount of collagen increases, leading to the hardening of the arteries [79]. The reduction in the elasticity of the tunica media is the result of reduced elastin expression and changes in elastin cross-linking and elastolysis [80]. The elastin is particularly susceptible to proteolytic degradation in atherosclerosis. Enzymes such as elastases, cathepsins, and MMPs degrade elastin fibers, leading to increased stiffness of the arteries and the formation of calcified plaques [81,82]. In addition, elastin degradation products, known as elastokines, can act as bioactive molecules that promote VSMC migration and proliferation [83,84].

### 4.3. Proteoglycans

Proteoglycans (PGs) consist of a protein core that is covalently bound to one or more glycosaminoglycan (GAG) chains [85,86]. GAGs are linear, negatively charged polysaccharides consisting of repeating disaccharide units [87], and are classified into five distinct categories: chondroitin sulphate (CS), dermatan sulphate (DS), heparan sulphate (HS), hyaluronan or hyaluronic acid (HA), and keratan sulphate (KS) [88]. In contrast to other GAGs, HA is not covalently bound to a nuclear protein and unsulphated [89]. Proteoglycans provide the vascular system with viscoelastic properties that are essential for maintaining proper vascular recovery and pulse vascular recovery and pulse wave propagation. Hyaluronan plays a crucial role in this process [18,90,91].

The endothelial glycocalyx, a specialized extracellular matrix on the luminal side of endothelial cells, acts as a barrier involved in cell adhesion, signal transduction, and the regulation of vascular permeability, thrombosis, and leukocyte extravasation [92,93]. The glycocalyx, which consists mainly of proteoglycans, GAGs, glycoproteins [94], and plasma proteins [95,96], varies between 0.5 and 5.0 μm [97,98,99,100] in the different vascular beds. Damage to the glycocalyx, which often occurs in vascular diseases, leads to increased permeability, allowing molecules from the plasma to enter the surrounding tissue [101,102].

Proteoglycans and GAGs are enriched in atherosclerosis and play a key role in plaque development [103,104,105,106]. GAGs modulate platelet aggregation, anticoagulation, and form complexes with low-density lipoproteins (LDLs) [107,108]. Versican, a large chondroitin sulphate proteoglycan, is the most abundant ECM proteoglycan in vessel walls and increases after vascular injury, accumulating in advanced plaques [105,109,110,111,112,113,114]. Versican promotes VSMC proliferation and migration, lipid storage, and plaque integrity [115,116]. However, its excessive accumulation can increase inflammation and destabilize plaques. In addition, hyaluronan fragments generated during ECM degradation act as DAMPs, promoting inflammation and further exacerbating atherosclerosis [117]. Biglycan and decorin, small leucine-rich proteoglycans (SLRPs), are upregulated in atherosclerosis, where they bind and trap LDL in the arterial wall. This promotes the oxidative modification of LDL and the formation of foam cells. They also interact with collagen fibrils, influencing ECM stability and plaque progression [38,103,118,119].

### 4.4. Fibronectin

Fibronectin is a multifunctional glycoprotein that plays a key role in normal tissue remodeling and development by affecting cell adhesion, migration, and wound healing [120,121,122]. Fibronectin is present in a soluble protomeric form in the blood plasma and in an insoluble multimeric form in the ECM. Insoluble fibronectin interacts with cells via integrins, heterodimeric transmembrane receptors that connect the ECM to the intracellular cytoskeleton, and signaling pathways [123,124]. During atherosclerosis, fibronectin expression is upregulated, particularly in areas of endothelial injury and inflammation [125,126]. This interaction promotes the recruitment of inflammatory cells in the plaque and supports VSMC migration and proliferation [127,128]. The presence of fibronectin in atherosclerotic plaques is associated with increased plaque instability, as it contributes to the formation of a disorganized and pro-inflammatory ECM [129,130].

### 4.5. Cartilage Oligomeric Matrix Protein

The cartilage oligomeric matrix protein (COMP) is a non-collagenous ECM protein known primarily for its role in the structural organization of cartilage. However, recent research has shown that COMP also plays an important role in the vascular system, particularly in the context of atherosclerosis. COMP is highly expressed in vascular SMCs, where it may help to maintain SMCs in a contractile state, which is associated with a more stable plaque phenotype. Conversely, the downregulation or dysfunction of COMP may promote the transition of SMCs to a synthetic, more proliferative state that contributes to plaque growth and instability [131]. In addition, COMP interacts with other ECM proteins such as collagen and fibronectin and regulates the production of inflammatory cytokines and matrix metalloproteinases (MMPs), which contributes to the organization and stability of the extracellular matrix and supports the structural integrity of the fibrous cap [132].

### 4.6. Fibrin

An important target for the detection of atherosclerotic plaques is fibrin, as it is not a significant protein in healthy vessel walls. Fibrin is a major component of the thrombi formed on the surface of atherosclerotic plaques. It plays a crucial role in plaque progression, the development of intraplaque haemorrhage, and thrombus formation after plaque rupture [133]. Fibrin accumulates in the ECM as a result of increased endothelial permeability during plaque development, with an increased number of blood components from newly formed vessels in early and advanced lesions [116,117].

### 4.7. Laminins

Laminins are a main component of the basement membrane and are involved in cell adhesion, differentiation, and migration [134,135]. They interact with cell surface receptors, such as integrins and dystroglycan, to regulate cellular functions and maintain tissue integrity [136,137]. In the early stages of atherosclerosis, alterations in laminin expression and structure can disrupt endothelial cell function, leading to increased permeability and leukocyte adhesion [138,139]. Laminin degradation by MMPs and other proteases further impairs the integrity of the basement membrane and promotes endothelial cell detachment and plaque instability. In addition, laminin fragments generated during ECM remodeling can influence the behavior of immune cells and contribute to the inflammatory milieu within the plaque [140,141].

## 5. ECM Remodeling in Atherosclerosis

ECM remodeling is a hallmark of atherosclerosis and involves both the synthesis of new matrix components and the degradation of existing components. This remodeling is driven by a variety of enzymes, including MMPs, cathepsins, and serine proteases, which are regulated by inflammatory cytokines and growth factors [142].

MMPs are considered the key enzymes responsible for the turnover of the ECM through the proteolytic degradation of virtually all of its proteinaceous components [143]. MMPs are a family of zinc-dependent endoproteases responsible for tissue remodeling and the degradation of ECM components, including collagens, elastin, and proteoglycans [144]. MMPs are produced by a variety of cell types in atherosclerotic plaques, including macrophages, VSMCs, and endothelial cells [145]. In atherosclerosis, MMPs are upregulated in response to inflammatory cytokines, such as TNF-α, IL-1β, and IFN-γ. MMP-1, MMP-2, MMP-3, MMP-9, and MMP-12 are particularly important for the degradation of the ECM in atherosclerotic plaques. As MMPs degrade the ECM, the migration and proliferation of VSMCs are accelerated [146]. MMP-8 has been detected in the shoulder regions of unstable atherosclerotic plaques, and like MMP-1, correlates with areas of cleaved collagen [147]. It has also been shown that MMP-8 is involved in the activation of other MMPs, such as MMP-2 and MMP-9 [148,149]. Increased expression of MMP-2 and MMP-9, known as gelatinases, degrade collagen IV and elastin, contributing to the weakening of the fibrous cap and increasing the risk of plaque rupture [150]. MMP-9 levels are positively related to the size of the necrotic core of coronary atherosclerotic plaques [151]. In contrast, MMP-12, which is mainly produced by macrophages, degrades elastin and promotes plaque calcification and instability [152].

Tissue inhibitors of metalloproteinases (TIMPs) are natural inhibitors of MMPs and play a crucial role in the regulation of ECM turnover [153]. In atherosclerosis, an imbalance between MMPs and TIMPs is frequently observed, with excessive MMP activity leading to pathological ECM degradation. This imbalance is an important factor in the development of unstable plaques [154,155].

Cathepsins are a family of cysteine proteases that are involved in the degradation of ECM components, in particular elastin and collagen. The activity of cathepsins is regulated by cystatin C, a potent inhibitor of cysteine proteases. In atherosclerosis, however, the balance between cathepsins and cystatin C is often disturbed, leading to the excessive degradation of the ECM [156]. Cathepsins are produced by macrophages and VSMCs and are upregulated in atherosclerotic plaques [157,158]. Cathepsins S, K, and L are particularly important in atherosclerosis. Cathepsin S is highly expressed in macrophage-rich areas of plaques and is able to degrade elastin, collagen, and proteoglycans. The activity of cathepsin S is associated with increased plaque instability and rupture [159,160]. Cathepsin K, originally identified in osteoclasts, is also expressed in VSMCs and macrophages within plaques, where it degrades collagen and contributes to plaque progression [161,162]. Cathepsins L has been reported to be significantly associated with apoptosis and plaque destabilization [163]. Serine proteases, including plasmin and neutrophil elastase, also play a role in ECM remodeling during atherosclerosis. The activity of serine proteases is regulated by serine protease inhibitors (serpins), but in atherosclerosis, the balance between proteases and inhibitors is often shifted towards proteolysis, leading to excessive ECM remodeling [164]. Plasminogen activators convert plasminogen to plasmin, which degrades fibrin and other ECM components [165]. Plasminogen activator inhibitor-1 (PAI-1) is the primary inhibitor of the tissue plasminogen activator (tPA) and urokinase plasminogen activator (uPA), and plays a crucial role in the regulation of fibrinolysis [166]. In atherosclerosis, elevated PAI-1 levels are associated with increased thrombosis and plaque instability [167,168]. Neutrophil elastase, which is produced by activated neutrophils, degrades elastin and collagen, and thus contributes to plaque instability [169]. Neutrophil elastase is upregulated under inflammatory conditions and contributes to ECM degradation by cleaving elastin and other matrix proteins [170,171].

Recently, next-generation sequencing (NGS) technologies have also shed light on the involvement of non-coding RNAs, such as microRNAs (miRNAs) and long non-coding RNAs (lncRNAs), which modulate ECM-related gene expression and contribute to the regulation of plaque stability [172,173,174]. Using RNA sequencing (RNA-seq), researchers have identified key genes and signaling pathways involved in ECM turnover. Recent studies have emphasized the role of MMPs and TIMPs in regulating ECM degradation and synthesis [175].

In addition, spatial transcriptomic (ST) analysis has allowed mapping the expression of ECM-related genes in different regions of atherosclerotic plaques, revealing spatial heterogeneity in the ECM composition [176]. Specifically, these studies have led to the discovery of distinct ECM signatures associated with vulnerable versus stable plaques. For instance, regions of high ECM remodeling, characterized by the increased expression of collagen-degrading enzymes such as MMP-9, were found in plaques with a higher risk of rupture. Unique cellular niches within plaques have been identified where the interactions between macrophages, smooth muscle cells, and endothelial cells cause localized changes in the ECM [177,178].

## 6. Therapeutic Implications

Given the central role of ECM proteins and their remodeling in the pathogenesis of atherosclerosis, targeting ECM components and the enzymes involved in their turnover represents a promising therapeutic strategy. Several approaches have been explored, including the following:

(a) MMP Inhibitors. The inhibition of MMP activity is considered a strategy to prevent plaque rupture. The inhibition of these enzymes may prevent the excessive degradation of collagen and thus preserve the strength of the cap [179]. For example, small-molecule inhibitors or monoclonal antibodies that selectively inhibit specific MMPs involved in plaque rupture could be developed to target these enzymes locally in atherosclerotic lesions. Early clinical trials using broad-spectrum MMP inhibitors, such as batimastat and marimastat, aimed to reduce MMP activity in various tissues [180]. However, these trials largely failed due to off-target effects, including musculoskeletal complications and impaired tissue repair. MMPs are involved in a wide range of physiological processes beyond ECM degradation, including embryonic development, wound healing, and immune response. As a result, the non-specific inhibition of these enzymes caused systemic side effects, limiting the clinical utility of early-generation MMP inhibitors [181,182]. More selective MMP inhibitors or inhibitors targeting specific MMPs, such as MMP-9, may offer a better therapeutic approach.

The focus has now shifted to the development of more selective inhibitors targeting specific MMPs involved in plaque rupture, such as MMP-9. By selectively inhibiting MMP-9, it may be possible to prevent collagen degradation in plaques without affecting other MMPs important for tissue homeostasis [183]. Localized delivery methods, such as drug-eluting stents or targeted nanoparticle-based systems, are also being explored to minimize systemic exposure and reduce side effects. With these approaches, MMP inhibitors can be delivered directly into atherosclerotic lesions, increasing therapeutic efficacy while limiting off-target effects [184,185]. Gene therapy approaches are currently being explored to locally increase TIMP expression in atherosclerotic plaques. By increasing TIMP levels, gene therapy could prevent excessive ECM degradation [186]. This targeted approach could improve plaque stability without the systemic side effects seen with pharmacological MMP inhibitors. In addition to gene therapy, synthetic TIMP analogs are currently being developed to mimic the activity of natural TIMPs [187]. These synthetic molecules could be designed to specifically inhibit MMP activity in plaques, restore ECM homeostasis, and reduce plaque instability.

Although selective MMP inhibition is promising, the complex role of MMPs in vascular biology and inflammation means that achieving a precise therapeutic window remains a challenge. In addition, the long-term suppression of MMP activity could interfere with other physiological processes, such as tissue regeneration, so a fine balance between therapeutic efficacy and safety is required.

(b) Promotion of ECM Stability. Therapeutic strategies aimed at restoring the balance of ECM synthesis and degradation are currently being investigated.

One possible strategy is the use of elastase inhibitors to prevent the degradation of elastin by enzymes such as neutrophil elastase. Agents such as sivelestat, a neutrophil elastase inhibitor, have been shown to protect elastin in preclinical models of vascular disease [188,189,190]. By reducing elastin degradation, these therapies may help to maintain the mechanical integrity of the arterial wall and thus reduce the risk of rupture. Another approach is to promote new elastin, inducing its synthesis. This could be achieved through targeted interventions in the metabolic pathways involved in elastin production. Elastin cross-linking agents such as β-aminopropionitrile may also help by reinforcing the structural integrity of existing elastin fibers, reducing vascular stiffness [191]. Drugs that enhance lysyl oxidase activity or stimulate elastin gene expression could promote the regeneration of damaged elastic fibers [192,193,194].

Strategies to promote collagen synthesis or inhibit collagen degradation are currently being researched. For example, drugs that modulate TGF-β signaling could increase collagen production by VSMCs, stabilize the fibrous cap, and reduce the risk of plaque rupture. Therapies targeting cathepsin or serine protease activity may help to maintain ECM integrity [195,196].

Promoting ECM stability by modulating elastin and collagen is promising. However, the excessive stimulation of ECM synthesis carries risks, such as pathological fibrosis [197]. In addition, the chronic inhibition of proteases could lead to the undesirable scarring or stiffening of tissue, which requires careful dosing and monitoring in clinical applications.

(c) Modulation of Proteoglycan–Lipoprotein Interactions. Another potential therapeutic strategy is to reduce lipoprotein retention in the arterial wall by interactions between targeting proteoglycan and lipoprotein [198,199]. This could be achieved by altering the structure or expression of proteoglycans, such as biglycan and decorin, or by using drugs that prevent the binding of LDL to proteoglycans.

Modifying the GAG chains bound to these proteoglycans could reduce their affinity for LDL and thus decrease lipid accumulation in the arterial wall. Gene-editing technologies such as CRISPR-Cas9 offer a potential method to selectively modify proteoglycan genes to reduce their ability to retain lipoproteins [200]. Another strategy is to develop small molecules or monoclonal antibodies that block the interaction between LDL and proteoglycans. By preventing the binding of LDL to the arterial ECM, these therapies could reduce lipid deposition and foam cell formation, thus slowing down plaque progression [201].

Therefore, targeting proteoglycans without impairing their normal physiological functions is a major challenge. Furthermore, modifying proteoglycan–lipoprotein interactions may not completely prevent the progression of atherosclerosis, as other mechanisms of lipid retention and oxidation may still contribute to plaque formation.

(d) Anti-inflammatory Therapies. Given the role that inflammation plays in ECM remodeling, anti-inflammatory therapies may also have beneficial effects on ECM turnover and plaque stability [202]. One of the most promising anti-inflammatory therapies is on the use of IL-1β inhibitors, such as canakinumab. In the CANTOS trial, canakinumab significantly reduced the rate of recurrent cardiovascular events in patients with atherosclerosis by inhibiting IL-1β. By targeting this signaling pathway, canakinumab not only decreases inflammation but may also help to stabilize the ECM within plaques, thereby reducing the risk of rupture [203,204].

Another novel approach involves targeting ECM fragments that are produced during degradation and may act as pro-inflammatory signals. These fragments, also known as matrikines, can recruit immune cells into the plaque, and thus exacerbate inflammation [205]. Neutralizing antibodies that block matrikines’ activity could reduce the recruitment of immune cells and create a less inflammatory environment in the plaque.

Anti-inflammatory therapies such as IL-1β inhibitors, while promising, also carry risks. The long-term suppression of the immune system could increase the risk of infections and other immune-related complications. Therefore, careful selection and monitoring of patients is crucial for the long-term efficacy of these treatments.

(e) Regulation of ECM-Associated Signaling Pathways. Targeting signaling pathways associated with ECM remodeling, such as the integrin signaling pathway, could provide another therapeutic approach. Inhibitors of specific integrins, such as αVβ3, have shown promise in preclinical studies in reducing plaque progression and instability [206,207]. Integrin inhibitors could potentially be used to modulate ECM turnover by preventing excessive degradation and promoting ECM synthesis, thereby stabilizing the plaque.

Interfering with ECM-associated signaling pathways is a complex process, due to the interplay between multiple molecular pathways that regulate ECM turnover. Furthermore, gene therapy approaches face significant technical and safety hurdles, including effective delivery to the target tissue and the potential for off-target effects.

## 7. Conclusions

The ECM is a central player in the pathogenesis of atherosclerosis and influences processes ranging from lipid retention to the recruitment of immune cells and plaque stability. The dynamic remodeling of the ECM is a key factor in the progression of atherosclerotic plaques and their propensity to rupture. Understanding the complex interactions between the ECM components, enzymes, and cells in the vessel walls is essential for the development of targeted therapies to stabilize plaques and prevent fatal cardiovascular events. As research continues to explore the mechanisms of ECM regulation in atherosclerosis, novel therapeutic strategies targeting ECM proteins and their remodeling may offer significant benefits in the management of this chronic and life-threatening disease. By stabilizing the ECM, reducing inflammation, controlling vascular stiffness, and intervening early in the disease process, these therapies could significantly decrease the risk of plaque rupture and associated cardiovascular events. Future therapies may involve combining strategies targeting the ECM with existing lipid-lowering therapies (e.g., statins or PCSK9 inhibitors) or anti-inflammatory drugs to provide a more comprehensive approach to treating atherosclerosis and improving cardiovascular outcomes. With advances in drug delivery systems, gene therapy, and biomarker development, ECM-targeted therapies hold promise for reducing the burden of atherosclerotic cardiovascular disease.

## Figures and Tables

**Figure 1 ijms-25-12017-f001:**
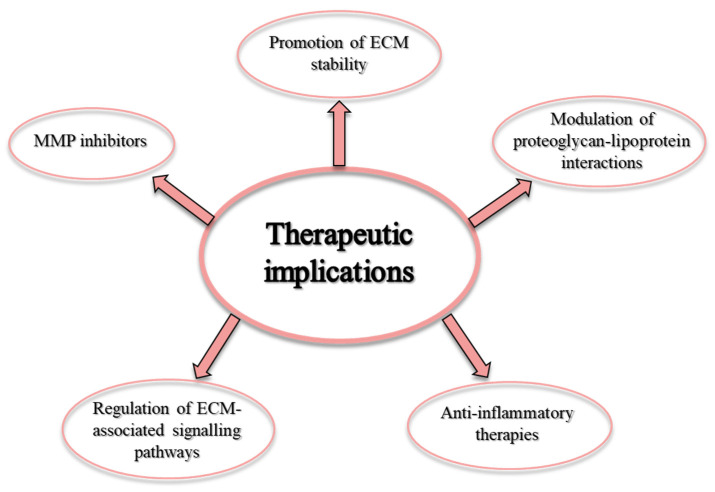
Therapeutic potential of ECM for atherosclerosis treatment.

**Figure 2 ijms-25-12017-f002:**
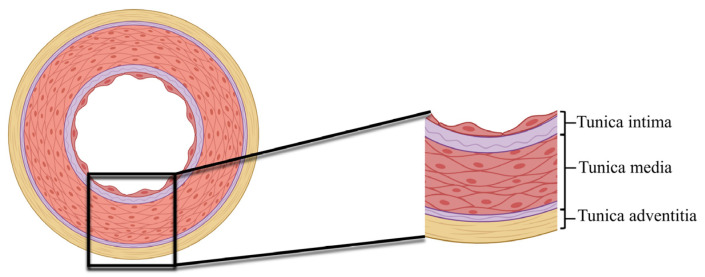
Cross-section of arterial ECM showing three layers: tunica intima, media, adventitia. Created with BioRender (2024).

**Figure 3 ijms-25-12017-f003:**
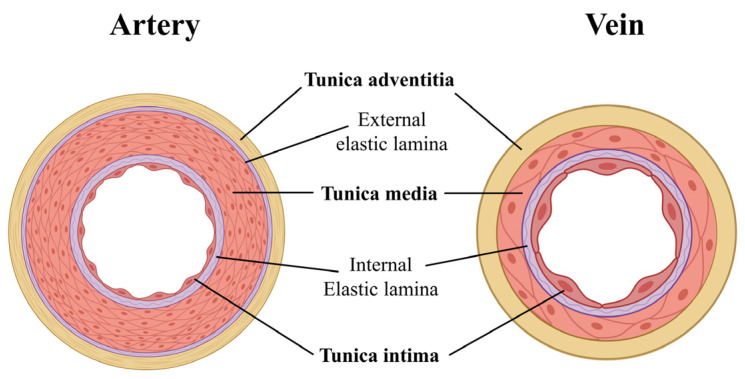
Differences in the structure of the ECM in arteries and veins. Created with BioRender.

**Figure 4 ijms-25-12017-f004:**
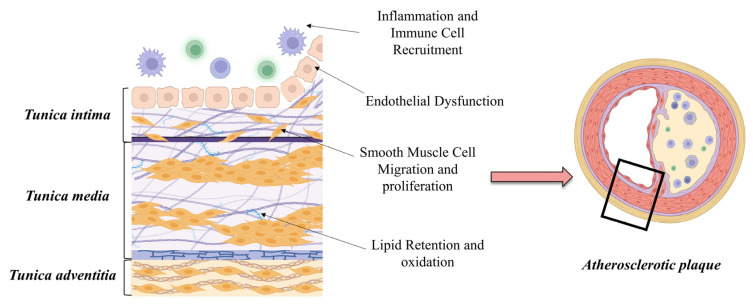
Schematic representation of ECM highlighting key processes influenced by atherosclerosis. Created with BioRender.

**Table 1 ijms-25-12017-t001:** Components of the three layers of the ECM.

Tunica Intima	Tunica Media	Tunica Adventitia
Collagen IV, VI, VIII, XV, XVIII	Collagen I, III, IV, V, VI	Collagen I, III, IV, V, VI
Fibronectin	Fibronectin	Fibronectin
Laminin	Laminin	Laminin
Biglycan	Proteoglycan	Proteoglycan
Perlecan	Decorin	Decorin
Versican	Elastin	Elastin

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
