# Peer review of "Vascular Extracellular Matrix in Atherosclerosis"

_ijms, 2024, doi:10.3390/ijms252212017_

Round 1
Reviewer 1 Report
Comments and Suggestions for Authors
In this manuscript authors discussed the structure, function of vascular extracellular matrix, focusing on its role and remodelling in atherosclerosis. Authors also evaluated ECM as potential therapeutic target for atherosclerosis.
the manuscript is interesting and the topic treated is clinically important. However, several points deserve improvements. In particular:
Abstract: authors must state the aim of the review
Lines 43-47: references are needed
Figure 2: although it is evident which one of the 2 images is the artery and which one is the vein, these should be indicated in the figure since the article could be read by non-expert people
Lines 274-275: it deserves to be highlighted that this plasticity in fibronectin plays a key role also in normal tissue remodeling and development (see PMID: 28076935)
A figure summarizing the main topics teeated in the manuscript would be helpful
An accurate revision of syntax is recommended
Author Response
Reviewer #1:
We have carefully read the Reviewer comments and we appreciate his valuable comments. We have addressed the different issues they brought up in detail. We believe that we have responded satisfactorily to all the points raised by reviewers. Our comments to each question are listed below.
Comment No 1: ‘Abstract: authors must state the aim of the review.’
-We thank the reviewer for finding this oversight. We have added the information.
Page 1, lines 16-17: “This review illustrates structure and role of vascular ECM, presenting a new perspective on ECM remodelling and its potential as therapeutic target in atherosclerosis treatments.
Comment No 2: ‘Lines 43-47: References are needed’.
- We added the missing references at page 2 line 50.
Comment No 3: ‘Although it is evident which one of the 2 images is the artery and which one is the vein, these should be indicated in the figure since the article could be read by non-expert people’.
- We thank the reviewer for their suggestion and revised Figure 2, now Figure 3 (page 3), accordingly.
Comment No 4: ‘Line 274-275: it deserves to be highlighted that this plasticity in fibronectin plays a key role also in normal tissue remodelling and development (see PMID:28076935)’
- As correctly suggested by reviewer we added this information at page 8 line 297-298.
Comment No 5: ‘Figure summarising the main topics treated in the manuscript would be helpful’.
-We thank the reviewer for their suggestion, and we created a new figure (Figure 1), page 2.
Comment No 6: ‘Revision of syntax is recommended.’
- We thank the reviewer for raising an important point. We have carefully revised and edit the manuscript.

Reviewer 2 Report
Comments and Suggestions for Authors
The Authors have summarized the components of vascular ECM and its central role in the pathogenesis of atherosclerosis. Some concerns are:
1. Most of the content, including the figures has not brought in further knowledge that has already been published in many reviews on the topic.
2. Sugession to introduce some recent discoveries based on next-generation sequencing and spatial transcriptome studies on ECM and atherosclerosis.
3. I suggest elaborating the phenotypic transition of different vascular cells and ECM.
4. To fill the gap, better to shorten the first 3 parts, but extend and explain more on the implications for therapeutic purposes.
5. The image of a vein in Figure 2 doesn’t show “thicker adventitia” as mentioned in the text.
6. Fig 3: Why is the long blood vessel in the layer of media?
Author Response
Reviewer 2:
We have carefully read the reviewer’ comments and we appreciate thevaluable comments from reviewer. We have addressed the different issues they brought up in detail. We believe that we have responded satisfactorily to all the points raised by reviewers. Our comments to each question are listed below.
Comment No 1: ‘Most of the content, including the figures has not brought in further knowledge that has already been published in many reviews on the topic.’
- We thank the reviewer for their insight. While it is true that some of the content and figures are consistent with existing reviews on this topic, our intention was to provide a comprehensive overview by combining existing knowledge with a new perspective on ECM remodelling in atherosclerosis and the ECM as a potential therapeutic target for clinical applications. We appreciate your observation and have added new insights, as you suggested in the comments below, to further distinguish this work from previous publications.
Comment No 2: ‘Suggestion to introduce some recent discoveries based on next-generation sequencing and spatial transcriptome studies on ECM and atherosclerosis.’
- We thank the reviewer for these interesting improvements. Thus, we added new paragraphs to the manuscript.
Page 10, lines 391-407: ‘Recently, next-generation sequencing (NGS) technologies have also shed light on the involvement of non-coding RNAs, such as microRNAs (miRNAs) and long non-coding RNAs (lncRNAs), which modulate ECM-related gene expression and con-tribute to the regulation of plaque stability [172-174]. Using RNA sequencing (RNA-seq), researchers have identified key genes and signalling pathways involved in ECM turnover. Recent studies have emphasised the role of MMPs and TIMPs in regu-lating ECM degradation and synthesis [175].
In addition, spatial transcriptomic (ST) analysis has allowed mapping the expres-sion of ECM-related genes in different regions of atherosclerotic plaques, revealing spatial heterogeneity in ECM composition [176]. Specifically, these studies have led the discovery of distinct ECM signatures associated with vulnerable versus stable plaques. For instance, regions of high ECM remodelling, characterised by increased expression of collagen-degrading enzymes such as MMP-9, were found in plaques with a higher risk of rupture. Unique cellular niches within plaques have been identified where in-teractions between macrophages, smooth muscle cells and endothelial cells cause lo-calised changes in the ECM [177, 178].’
Comment No 3: ‘I suggest elaborating the phenotypic transition of different vascular cells and ECM.’
- We thank the reviewer for their useful suggestions. Therefore, we added the necessary information in the manuscript.
Pages 4-5, lines 130-142: ‘ECs can undergo an endothelial-to-mesenchymal transition (EndMT), a dedifferentiation process in which they lose their endothelial characteristics (e.g., reduced expression of endothelial markers such as VE-cadherin, endothelial nitric oxide synthase (eNOS)) [29,30] and gain mesenchymal features (e.g., increased expression of mesenchymal markers such as alpha-smooth muscle actin (α-SMA), vimentin, N-cadherin) [31]. During EndMT, disruption of endothelial junctions allows high amounts of inflammatory cells, lipids, and immune complexes to penetrate the intima, which further exacerbates disruption of ECM components like laminin and collagen IV and promotes plaque growth [32,33]. In addition, degradation of ECM components, particularly by matrix metalloproteinases (MMPs), can release matrix-bound growth factors, such as transforming growth factor-β (TGF-β), which promotes EndMT and enhances the fibrotic and inflammatory environment within the plaque [34].’
Page 5,lines 158-168: ‘This synthetic form is characterized by increased proliferation, migration, and as reduced contractile protein expression, as well ECM production [41]. Synthetic VSMCs produce ECM components such as collagen and elastin, which can contribute to plaque stabilization. However, their ability to secrete MMPs can degrade the ECM, leading to plaque destabilization and vulnerability to rupture [42].
Interestingly, changes in ECM stiffness due to collagen accumulation, elastin degradation or other ECM components, can influence the phenotypic transition of smooth muscle cells (SMCs) [42]. Increased ECM stiffness is associated with the promotion of the synthetic phenotype of VSMCs, which leads to excessive ECM production and contributes to plaque growth [42].’
Comment No 4: ‘To fill the gap, better to shorten the first 3 parts, but extend and explain more on the implications for therapeutic purposes.’
-We thank the reviewer for their helpful advice. Therefore, we edited the manuscript accordingly.
Pages 10-11, Lines 413-446: Inhibition of these enzymes may prevent excessive degradation of collagen and thus pre-serve the strength of the cap [179]. For example, small-molecule inhibitors or monoclonal antibodies that selectively inhibit specific MMPs involved in plaque rupture could be developed to target these enzymes locally in atherosclerotic lesions. Early clinical trials using broad-spectrum MMP inhibitors, such as batimastat and marimastat, aimed to reduce MMP activity in various tissues [180]. However, these trials largely failed due to off-target effects, including musculoskeletal complications and impaired tissue repair. MMPs are involved in a wide range of physiological processes beyond ECM degradation, including embryonic development, wound healing, and immune response. As a result, non-specific inhibition of these enzymes caused systemic side effects, limiting the clinical utility of early-generation MMP inhibitors [181, 182].
Line 426-459: The focus has now shifted to the development of more selective inhibitors targeting specific MMPs involved in plaque rupture, such as MMP-9. By selectively inhibiting MMP-9, it may be possible to prevent collagen degradation in plaques without affecting other MMPs important for tissue homeostasis [183]. Localised delivery methods, such as drug-eluting stents or targeted nanoparticle-based systems, are also being explored to minimise systemic exposure and reduce side effects. With these approaches, MMP inhibitors can delivered directly into atherosclerotic lesions, increasing therapeutic efficacy while limiting off-target effects [184, 185]. Gene therapy approaches are currently being explored to locally increase TIMP expression in atherosclerotic plaques. By increasing TIMP levels, gene therapy could prevent excessive ECM degradation [186]. This targeted approach could improve plaque stability without the systemic side effects seen with pharmacological MMP inhibitors. In addition to gene therapy, synthetic TIMP analogues are currently being developed to mimic the activity of natural TIMPs [187]. These synthetic molecules could be designed to specifically inhibit MMP activity in plaques, restore ECM homeostasis and reduce plaque instability.
Although selective MMP inhibition is promising, the complex role of MMPs in vascular biology and inflammation means that achieving a precise therapeutic window remains a challenge. In addition, long-term suppression of MMP activity could interfere with other physiological processes, such as tissue regeneration, so a fine balance between therapeutic efficacy and safety is required.
Page 11, Lines 447-459: b) Promotion of ECM Stability. Therapeutic strategies aimed at restoring the balance of ECM synthesis and degradation are currently being investigated.
One possible strategy is the use of elastase inhibitors to prevent the degradation of elastin by enzymes such as neutrophil elastase. Agents such as sivelestat, a neutrophil elastase inhibitor, have been shown to protect elastin in preclinical models of vascular disease [188-190]. By reducing elastin degradation, these therapies may help to maintain the mechanical integrity of the arterial wall and thus reduce the risk of rupture. Another approach is to promote new elastin inducing its synthesis. This could be achieved through targeted interventions in the metabolic pathways involved in elastin production. Elastin cross-linking agents such as β-aminopropionitrile may also help by reinforcing the structural integrity of existing elastin fibres, reducing vascular stiffness [191]. Drugs that enhance lysyl oxidase activity or stimulate elastin gene expression could promote the regeneration of damaged elastic fibres [192-194].
Page 11, lines 465-468: Promoting ECM stability by modulating elastin and collagen is promising. However excessive stimulation of ECM synthesis carries risks such as pathological fibrosis [196]. In addition, chronic inhibition of proteases could lead to undesirable scarring or stiffening of tissue, which requires careful dosing and monitoring in clinical applications.
Page 11, lines 474-485: Modifying the GAG chains bound to these proteoglycans could reduce their affinity for LDL and thus decrease lipid accumulation in the arterial wall. Gene-editing technologies such as CRISPR-Cas9 offer a potential method to selectively modify proteoglycan genes to reduce their ability to retain lipoproteins [200]. Another strategy is to develop small molecules or monoclonal antibodies that block the interaction between LDL and proteoglycans. By preventing binding of LDL to the arterial ECM, these therapies could reduce lipid deposition and foam cell formation, thus slowing down plaque progression [201].
Therefore, targeting proteoglycans without impairing their normal physiological functions is major challenge. Furthermore, modifying proteoglycan-lipoprotein interactions may not completely prevent the progression of atherosclerosis, as other mechanisms of lipid retention and oxidation may still contribute to plaque formation.
Pages 11-12, lines 489-503: One of the most promising anti-inflammatory therapies is on the use of IL-1β inhibi-tors, such as canakinumab. In the CANTOS trial, canakinumab significantly reduced the rate of recurrent cardiovascular events in patients with atherosclerosis by inhibit-ing IL-1β. By targeting this signalling pathway, canakinumab not only decreases in-flammation but may also help to stabilise ECM within plaques, thereby reducing the risk of rupture [203, 204].
Another novel approach involves is to target ECM fragments that are produced during degradation and may act as pro-inflammatory signals. These fragments, also known as matrikines, can recruit immune cells into the plaque, and thus exacerbate in-flammation [207]. Neutralising antibodies that block matrikines activity could reduce the recruitment of immune cells and create a less inflammatory environment in the plaque.
Anti-inflammatory therapies such as IL-1β inhibitors, while promising also carry risks. Long-term suppression of the immune system could increase the risk of infec-tions and other immune-related complications. Therefore, careful selection and moni-toring of patients is crucial for the long-term efficacy of these treatments.
Page 12, lines 508-513: Interfering with ECM-associated signalling pathways is a complex process, due to the interplay between multiple molecular pathways that regulate ECM turnover. Further-more, gene therapy approaches face significant technical and safety hurdles, including effective delivery to the target tissue and the potential for off-target effects.
Page 12, lines 523-531: By stabilising the ECM, reducing inflammation, controlling vascular stiffness and in-tervening early in the disease process, these therapies could significantly decrease the risk of plaque rupture and associated cardiovascular events. Future therapies may in-volve combining strategies -targeting the ECM with existing lipid-lowering therapies (e.g. statins or PCSK9 inhibitors) or anti-inflammatory drugs to provide a more com-prehensive approach to treating atherosclerosis and improving cardiovascular out-comes. With advances in drug delivery systems, gene therapy and biomarker devel-opment, ECM-targeted therapies hold promise for reducing the burden of atheroscle-rotic cardiovascular disease.
Comment No 5: ‘The image of a vein in Figure 2 doesn’t show “thicker adventitia” as mentioned in the text.’
-We thank the reviewer for detecting this oversight. We modified accordingly the figure (now Figure 3, page 3).
Comment No 6: ‘Fig 3: Why is the long blood vessel in the layer of media?’
- We thank the reviewer for finding this mistake. We edited completely the figure (now Figure 4, page 4).

Round 2
Reviewer 1 Report
Comments and Suggestions for Authors
the manuscript has been significantly improved and can be accepted in the present form